# Challenges and Opportunities for Cover Crop Mediated Soil Water Use Efficiency Enhancements in Temperate Rain-Fed Cropping Systems: A Review

**Eric Britt Moore** (ID)

Department of Environmental Sciences, University of North Carolina Wilmington, 1080A Dobo Hall, Wilmington, NC 28411, USA; mooree@uncw.edu

**Abstract:** Soils are at the nexus of the atmospheric, geological, and hydrologic cycles, providing invaluable ecosystem services associated with water provision. The immeasurably vital role of water provision is of urgent concern given the intertwined and interdependent challenges of growing human populations, increased agricultural demands, climate change, and freshwater scarcity. Adapting temperate rain-fed cropping systems to meet the challenges of the 21st century will require considerable advancements in our understanding of the interdependent biophysical processes governing carbon and soil-water dynamics. Soil carbon and water are inextricably linked, and agricultural management practices must take this complexity into account if crop productivity is to be maintained and improved. Given the widespread, intensive use of agricultural soils worldwide, it stands to reason that readily adaptable crop management practices can and must play a central role in both soil carbon and water management. This review details challenges and opportunities for utilizing cover crop management to enhance soil carbon stocks and soil water use efficiency in rain-fed cropping systems. A review of the current body of knowledge shows that cover crops can play a more prominent role in soil carbon and water management; however, the more widespread use of cover crops may be hindered by the inconsistencies of experimental data demonstrating cover crop effects on soil water retention, as well as cover crop effect inconsistencies arising from complex interactions between soil carbon, water, and land management. Although these gaps in our collective knowledge are not insignificant, they do present substantial opportunities for further research at both mechanistic and landscape-system scales.

**Keywords:** cover crops; rain-fed cropping systems; plant available water; soil organic carbon; water use efficiency



## 1. The Challenges Facing Soil and Water Resource Management

Agriculture is in the midst of a defining era. Whereas precise estimates vary, there is near universal consensus that global food production and sustainable soil management practices will need to increase substantially in the coming decades to meet the demands of a growing population [1]. These production gains must occur against the backdrop of climate change; and given the importance of climate to soil formation processes [2], climate change is likely to cause significant disruptions to global agricultural production [3]. Incidences of extreme temperatures and heatwaves are expected to increase significantly [4,5], likely straining rain-fed cropping systems. The availability of freshwater, the lifeblood of agriculture, is also expected to change markedly in the coming decades. Warmer air temperatures are expected to cause changes to global precipitation patterns, which will increase the severity of floods in some areas and exacerbate drought in others [6,7]. Additionally, trends in groundwater depletion, if continued, will further aggravate water stresses in regions across the globe [8,9].

The economic impacts of climate change are expected to have profound consequences for U.S. and global agricultural markets [10]. Environmental stresses associated with

climate change are expected to diminish U.S. agricultural production [11] and by extension the export of virtual water [9,12]. Significant reductions in the export of virtual water, which is the combined total of water needed in each step of the agricultural production process, is likely to exacerbate global food insecurity [12] particularly in arid and semi-arid regions [12,13].

The challenges that climate change present are compounded by the continuance of severe and pervasive degradation of agricultural soils, which presents an extraordinary challenge to sustaining and improving upon current levels of agricultural production. Across the globe, approximately 25–30% of all agricultural lands are classified as degraded [14,15], and soils are being degraded at a faster rate than they can be sustained [16]. Soils are sensitive to the impacts of agricultural production and can take decades, or longer, to recover pre-cultivation soil quality characteristics [17,18]. Given the fact that soil water supports approximately 90% of global agricultural production [19], soil degradation is inescapably tied to water management. Climate change impacts in temperate cropland regions will likely result in an intensification of excessive spring precipitation events [20], particularly in the U.S. Corn Belt, one of the highest producing rain-fed agriculture regions in the world. These impacts are expected to reduce overall crop yields and exacerbate soil erosion [21]. Moreover, crop yield reductions from these environmental stressors will not be fully compensated for by crop physiological benefits attributable to increased atmospheric carbon [13,22]. The cumulative effects of climate change and natural resource depletion, along with a burgeoning global middle class and a growing world population, will require agriculture to intensify while simultaneously using fewer resources. Failure to implement viable solutions to address these momentous challenges will increase the likelihood of conflict and political instability across the globe [23].

Adapting temperate rain-fed cropping systems to meet the challenges of the 21st century will require considerable advancements in our understanding of the interdependent biophysical processes governing carbon and soil-water dynamics. Soil carbon and water are inextricably linked, and agricultural management practices must take this complexity into account if crop productivity is to be maintained and improved. Sustainable intensification of agriculture is possible, and although big-data and improved cultivars will undoubtedly play a significant role in this transition, agriculture will fall short of the monumental challenges that lie ahead unless and until better soil carbon and water resource management strategies are developed. This challenge is demanding; however, viable solutions are attainable.

## 2. The Case for Cover Crops

One soil resource management strategy that has been gaining increasing popularity is the use of cover crops. Although the total percentage of farmland managed with cover crops remains minute, growth rates in cover crop adoption may be signaling a shift in the role that cover crops play in modern row-crop agriculture [24]. Cover crop acres have increased by an average of 50% in the U.S. between 2012 and 2017 [25]. Iowa, which leads the U.S. Corn Belt in cover crop gains, had approximately 3% of its farmland managed with cover crops in 2017; however, the increase in cover crop acreage was 156% from 2012 to 2017 [26,27]. Increases in cover crop adoption are due in large part to the demonstrated benefits for soil and nutrient conservation. For example, cover crops have been shown to aid in pest suppression [28], reduce soil erosion [29], mitigate nutrient leaching [30], and enhance soil organic matter content [31]. Cover crop cultivation on agricultural landscapes also has the potential to sequester carbon in soils at levels of significance for global greenhouse gas emissions management [8]. As living roots play a substantive role in shaping the soil biophysical environment [32,33], it stands to reason that cover-crop-mediated ecosystem services are attributable, at least in part, to enhanced root zone biological activity.

The importance of soil microbes in facilitating and maintaining soil biogeochemical processes cannot be overstated. In addition to mediating nutrient cycles [34], microbes are also an integral part of soil food webs, creating microhabitats that in turn foster enhanced

biodiversity in the root zone [35]. Soil microbes require carbon compounds to satisfy their energy demands, which make them highly dependent on plant root exudates. A temporal expansion of the rhizosphere (e.g., incorporation of winter cover crops), can improve soil quality through enhancing soil biological activity [36], which in turn promotes soil aggregation [37–39] and soil organic matter accrual [40,41]. Cover crops can increase soil carbon by more than double compared with treatments that added one or more crops to a rotation but which lack a cover crop [37], adding further evidence that the benefits of cover crops are due, in part, to a temporal expansion of the rhizosphere during the growing season.

Although cover crops have been shown to improve soil quality [28], relatively little is known about cover crop influences on soil water retention. Nichols et al. [42] found that long-term cover management did not affect soil macroporosity or water content at saturation, although there was some increase in water content at field capacity. Undisturbed native soils often have higher water content than cultivated soils, with restored soils having intermediate water content [43]. It seems, therefore, reasonable to infer that minimal soil disturbance, increased biodiversity, perennial surface cover, and perennial living roots alter soil-water dynamics; however, an empirical basis for extrapolating these results to enhancements in soil-water use efficiency has yet to be firmly established. A promising means by which to conserve freshwater resources in temperate climes is enhanced soil-water use efficiency arising from increased plant populations [16,44]; however, to date, relatively few studies have investigated the effects of cover crops on soil-water use efficiency in temperate rain-fed cropping systems.

## 3. Challenges and Opportunities for Cover Crops as a Soil Water Management Strategy

Plant available water (PAW) is defined as the amount of soil water that is held between field capacity (FC) and permanent wilting point (PWP). The standard used for FC varies across the globe, with values generally ranging from $-5$ to $-33$ kPa; however, a fixed value of $-1500$ kPa for PWP is more universally agreed upon [45]. Although water in soils is physically present at matric tensions less than $-1500$ kPa (i.e., PWP), the water is held so tightly to the soil matrix that plants are typically unable to extract it at a rate sufficient to meet the transpiration demand.

Estimations of PAW are dependent on accurate measurements of FC. Field capacity is the water content at which free drainage (i.e., drainage due to gravity) of a previously saturated soil has become negligible. The period between wetting and negligible drainage is usually 48–72 h [46]. Although $-33$ kPa is the generally accepted value for FC, there can be substantial variability across soils and landscapes [47].

One of the challenges that complicates the study of soil-water properties is an accurate approximation of the relative importance of soil organic matter (SOM) across varying matric potentials. Foundational research by Hudson [48] and Emerson [49], as well as more recent analyses such as Basche et al. [50] and Nichols et al. [42], reported increases in soil water retention resulting from organic carbon additions are more pronounced at FC than at PWP. Therefore, soil water retention characteristics could potentially be influenced by soil organic inputs derived from management practices that augment plant root biomass, such as cover crops. There remains, however, uncertainty of the degree to which cover crop management can alter soil water retention characteristics. Whereas cover crops can significantly increase soil water at FC [50], there can be substantial variation in the extent to which cover crops augment soil water in rain-fed systems [51]. For example, Nichols et al. [42] showed that some rain-fed cropping sites had significant increases in volumetric water content at FC as a result of cover crop management, whereas other comparable sites showed no cover crop treatment effect (Figure 1).

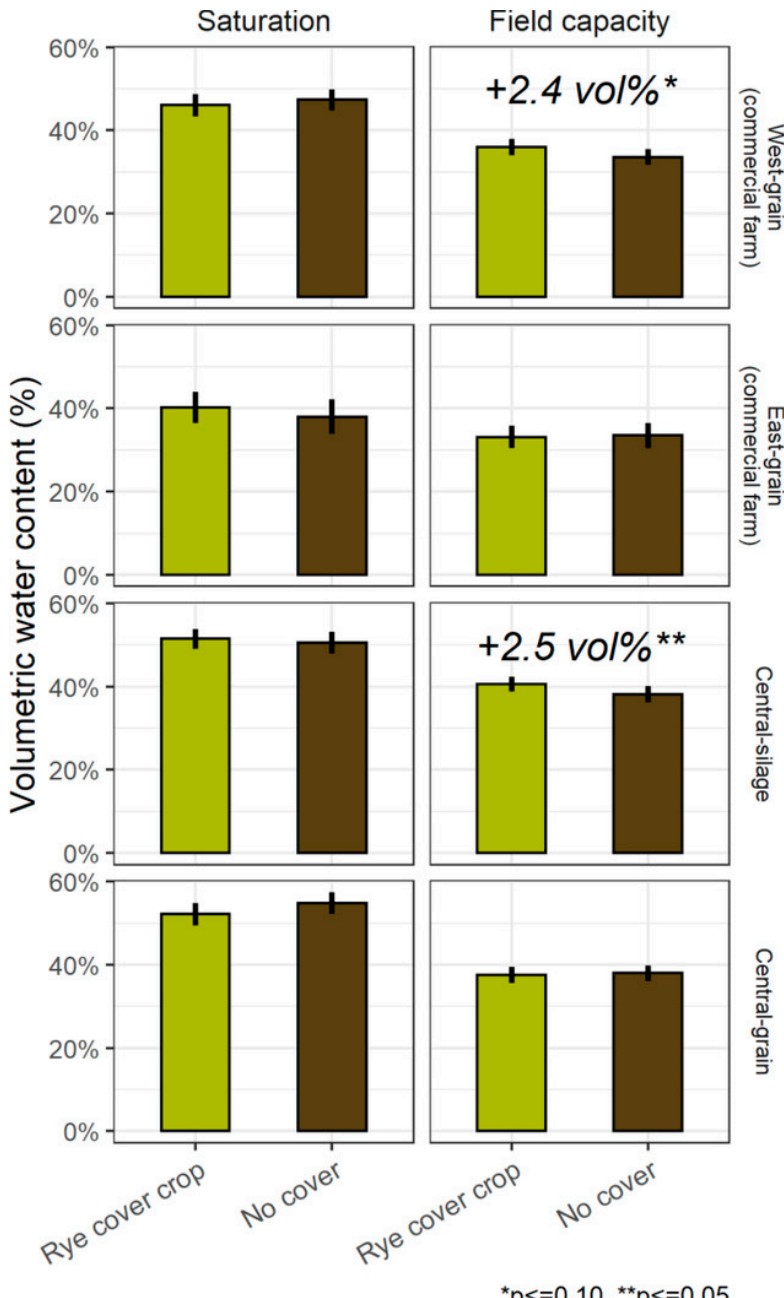

**Figure 1.** Soil volumetric water contents at saturation and field capacity (−9.8 kPa) with 10+ years of winter rye cover cropping (green) or winter fallow (brown) in a maize–soybean rotation at four trials. Bars show estimated means, line ranges are the 95% confidence intervals around the mean. Text presents the estimated effect of cover cropping on volumetric water content in instances of a significant effect. Data from [42]: Nichols, V.; Moore, E.B.; Gailans, S.; Kaspar, T.; Liebman, M. Site-specific effects of winter cover crops on soil water storage. *Agroecosystems Geosci. Environ.* **2022**, *5*, e220238.

Another factor that merits serious consideration is the soil structure changes caused by enhanced root zone activity. Evidence suggests that cover crop roots can alter the soil structure significantly [39]. These changes can be achieved, in part, through increasing the number of soil macropores [52], which in turn increases water infiltration [41]. Water flow and retention can also be influenced by hydrophobic compounds produced by the fungal hyphae [53,54]. There is also evidence to suggest that soil microbe-derived labile organic

matter has hydrophilic properties that enable it to retain disproportionately large volumes of water [48,55].

One of the factors that both complicates and speaks to the importance of soil carbon–water research is anthropogenic impacts on soil. Soil erosion rates, although improving in many areas, remain alarmingly high [56] and are occurring at a pace that is unsustainable for long-term productivity [57,58]. Soil erosion has altered landscapes across the globe, resulting in loss of topsoil and the concomitant loss of soil organic matter. Tillage, which is a major contributor to soil erosion, can alter the soil's structure-dependent properties as a function of both depth and time [59]. Increased soil erosion will most certainly strain sustained crop productivity, as topsoil contains an exceptionally large volume fraction of the total organic matter in the solum. Increased soil erosion, and the concomitant loss of SOM, could decrease plant available water [20,48], further exacerbating the crop stresses associated with climate change. It therefore stands to reason that soil hydrology has already been disproportionately impacted by the loss of topsoil across intensively managed agricultural landscapes.

Iowa, for example, has a well-documented history of anthropogenic soil formation and transformation processes over the past 50+ years [60] and has documented soil erosion rates that exceed soil formation by at least an order of magnitude [56,58]. The continued rate of unsustainable soil losses will undoubtedly alter soil hydrology, highlighting the urgency for implementation of management practices that effectively combat soil erosion. Analyses by Kaspar et al. [61], Wilhelm et al. [29], and Koudahe et al. [41] suggest that cover-crop-derived enhancements to soil quality offer a viable means to ameliorate the impacts of soil erosion.

Another major anthropogenic factor that may influence soil erosion rates, and concomitantly soil hydrology, is climate change. Climate models predict that the U.S. Corn Belt will experience changes in annual precipitation patterns that will likely result in a greater number of high-intensity rainfall events [12,20,62], increasing the risk of summer flooding. Improved flood and drought management strategies will be required to mitigate crop yield losses amidst a changing climate. The poorly drained soils that comprise significant areas of agricultural land in the U.S. Corn Belt make flooding concerns in the region acutely relevant to global food security.

Agricultural management practices that improve soil water retention may offer a means to blunt the severity of flooding. A foundational study by Emerson [49] found that increases in soil carbon are positively correlated with increases in soil water retention, with these increases more pronounced at field capacity than at permanent wilting point. More recent studies, including Basche et al. [50] and Nichols et al. [42], support these findings. These results suggest that agricultural practices that increase soil carbon could serve as viable components of integrated water management plans, particularly for extreme precipitation events. Additionally, cover crops may aid flood mitigation in tile-drained landscapes [63]. Although flooding results from a multitude of converging factors, including antecedent soil moisture, topography, drainage, and infrastructure design, a better understanding of soil water retention in the root zone would undoubtedly serve to strengthen integrated flood management strategies in temperate rain-fed cropping regions. Although there is urgent need to expand agricultural management tools to address sustainable intensification, there remains a dearth of basic scientific knowledge regarding the biophysical processes governing the interactions between texture, organic carbon, soil structure, and soil water retention.

## 4. The Role of Soil Texture in Water Retention

One of the basic soil properties that can help elucidate the impacts of cover crops on soil water retention is texture. Soil texture is a dominant factor in determining the kinetics of soil water transport and retention processes [64,65]. It is well established that coarser-textured soils have larger pore spaces and increased hydraulic conductivity relative to finer-textured soils; however, the relationship between soil texture and soil water retention

is complicated by climate and soil organic carbon (SOC) interactions. Climate is the sine qua non of soil formation, especially given its interdependence with other key factors such as biologic and physical weathering processes. Soil textural differences account for most of the soil-water variability under wet conditions, and SOC is responsible for most of the water variability under dry conditions, with the sand fraction and SOC fraction being the best predictors of soil water content across a wide range of soils [66].

The impacts of SOC seem to be most pronounced in coarse-textured soils, as evidenced by significant SOM alterations to soil matric potential in sandy soils [67,68]. Plant available water enhancements attributable to SOC inputs are also more pronounced in coarse-textured soils when compared with other texture classes [45,69], even when SOC is added to coarse-textured soils with relatively high antecedent SOC [70]. Furthermore, when SOC is added to coarse-textured soils with low antecedent SOC, soil water retention responds more dramatically to SOC additions, and although SOC additions to low SOC fine-textured soils decreases soil water retention, SOC additions at high SOC levels increases soil water retention across all soil textures [70].

Clay and silt content are significant factors in determining how SOC inputs respond to management changes, with clay–silt soils storing increased amounts of particulate organic matter (POM) after they have become carbon saturated [71]. Higher clay content generally results in reduced SOC losses in cultivated soils, and although SOC increases with precipitation in uncultivated soils, the response is the opposite in cultivated soils [72]. These foundational data lend credence to the theory that clay protects SOM against microbial degradation, indicating that clay content has an exceptional impact on SOC levels across landscapes.

## 5. The Role of Soil Organic Matter in Water Retention

Soil physical properties are significantly influenced by organic matter; however, soil organic matter is a broadly defined term; it technically encompasses everything from living organisms to decomposing plant tissues and humic substances. Soil organic matter pools vary significantly depending upon multiple factors, including age and chemical composition of the original organic substrate, soil hydrology, and climate [73]. Soil organic matter is often categorized according to relative chemical reactivity. These categories include labile, slow, and recalcitrant organic matter pools.

The labile SOM pool has a decomposition rate ranging from several months to several years and consists largely of microbes and microbial-derived substances [73]. One category of microbial-derived substances germane to soil carbon–water research is extracellular polymeric substances (EPS). These carbon-rich compounds are secreted by soil microbiota and are a widespread characteristic of soil microbes across multiple phylogenic lines [53,54]. Although EPS constitute a relatively small percentage of SOM, they have been shown to promote soil aggregation and ameliorate the impacts of rapid changes in soil water potential [53,74,75]. Extracellular polymeric substances have chemical properties that endow them with significant water storage capacity. A study by Rosenzwieg et al. [55] has demonstrated the ability of xanthan, an EPS analogue, to increase soil water content by as much as 270%. Although the previous study analyzed water retention resulting from the addition of isolated xanthan, it stands to reason that in situ EPS would elicit similar responses. The role of microbial EPS in soil-water dynamics is complicated by the fact that certain fungal EPS exhibit hydrophobic properties [53,75], which itself is unsurprising given that most soil fungi are obligate aerobes and therefore require protection against extended periods of saturation. Soil EPS seem to buffer against rapid changes in soil water potential that can cause hypoxia, cellular lysis, or pneumatic rupturing of soil aggregates. As such, soil EPS can function in both hydrophilic and hydrophobic capacities depending on factors such as the relative abundance of carbon-rich plant exudates and soil matric potential.

Measurements of EPS could potentially serve as a proxy for comparisons of water retention across various soils of similar textural classes, as small changes to EPS have been shown to result in significant changes in soil water retention. Labile carbon mineralization shows a strong relationship with soil microbial biomass carbon [76] and could serve as a proxy for changes in soil extracellular polymeric substances. Permanganate-oxidizable carbon (POXC) may also serve a similar function; however, evidence of its utility remains inconclusive. Although there is evidence to suggest that POXC is highly correlated with biomass carbon [77] and is appropriate for comparing management practices [78], POXC may not be a reliable measure of total labile carbon [79] or management-induced SOC changes under all circumstances [80], calling into question the reliability of this metric to compare data across studies.

The slow SOM pool has a decomposition rate ranging from 20 to 50 years and consists of plant structural compounds, such as lignin, that are relatively resistant to decomposition. Particulate organic matter has properties that are consistent with the slow SOM pool [81,82] and can serve as a sensitive indicator of soil quality changes resulting from crop management [31] (Table 1). Particulate organic matter shows evidence of acting as a transitionary space for soil organic matter that can either be utilized by microbes upon mineralization or transitioned into longer-term (i.e., recalcitrant) organic matter storage [82]. Fine-textured soils store increased amounts of POM after they have become carbon saturated, which suggests that POM can transition cyclically into active organic matter pools as ambient carbon availability changes [71]. Furthermore, long-term cover crop management can attribute up to 38% of SOM gains to POM increases [31], and significant increases in POM can occur even when total carbon storage increases are negligible [38].

**Table 1.** Particulate organic matter (POM) in the 0 to 5 cm and 5 to 10 cm depth layers for treatments with and without a rye cover crop in a corn silage–soybean rotation averaged over 2 years and in two adjacent fields.

| Depth Layer | 0- to 5-cm Depth | | | 5- to 10-cm Depth | | |
|---|---|---|---|---|---|---|
| **Previous Crop** [†] | **Corn** | **Soybean** | **Avg.** | **Corn** | **Soybean** | **Avg.** |
| | | | g POM kg soil$^{-1}$ | | | |
| Treatment | | | | | | |
| Rye after Silage | 8.8 a [‡] | 8.1 ab | 8.4 a | 3.7 a | 3.4 a | 3.5 a |
| Rye after Both | 8.8 a | 8.9 a | 8.8 a | 3.8 a | 4.3 a | 4.0 a |
| No Cover Crop | 6.3 b | 5.9 c | 6.1 b | 3.0 a | 3.3 a | 3.2 a |
| Rye after Soybean | 5.8 b | 6.7 ab | 6.3 b | 3.4 a | 3.2 a | 3.3 a |
| Previous Crop Avg. | 7.4 A | 7.4 A | | 3.5 A | 3.6 A | |

[†] Previous crop refers to main crop that was present in a field the year before soil samples were taken. [‡] Numbers within a column followed by the same lowercase letter and numbers within a row and depth layer followed by the same uppercase letter are not significantly different as indicated by LSD test at the 0.05 probability level. Data adapted from [31] Moore, E.B.; Kaspar, T.C.; Wiedenhoeft, M.H. Cambardella, C.A. Rye cover crop effects on soil properties in no-till corn-silage-soybean cropping systems. *Soil Sci. Soc. Am. J.* **2014**, *78*, 968–976.

The recalcitrant SOM pool has a decomposition rate of 400–2000 years and consists of refractory compounds including humic acids and biochar [73]. Recalcitrant SOM is generally deemed important for cation exchange capacity, especially in coarse and medium-textured soils [83]; however, the effects can be inconsistent [68]. Combined biochar and manure applications may actually serve to immobilize nutrients in medium-textured soils [84]. Recalcitrant SOM can also alter soil matric potential for given gravimetric water contents [85], particularly in sandy soils [86]. Although recalcitrant SOM plays an important, and complex, role in soil biogeochemical processes [83], edaphological studies that do not incorporate biochar amendments or employ controlled burning have little reason to suspect that baseline recalcitrant organic matter levels will change as a result of cover crop management.

Despite studies demonstrating the positive impacts of SOM on soil water retention, there remains a lack of consensus on the extent to which SOM alters water retention characteristics across the plant available water range. Although soil texture is the dominant factor in determining the kinetics of soil water transport and retention processes, SOM affects the shape and position of the soil water retention curve, resulting in higher water content across the PAW range [64]. Seminal research by Hudson [48] found a strong positive correlation between volumetric water content and SOM at field capacity in coarse- and medium-textured soils. Plants colonized by arbuscular mycorrhizal fungi can significantly increase soil water contents across all PAW matric potentials in soils where compost was also present [87]. However, increases in soil water content from SOM additions are sometimes negligible. An extensive analysis by Minasny & McBratney [45] found that, on average, a 1% increase in SOC corresponds to a 1–3% increase in soil volumetric water content. Meta-analytic data from Libohova et al. [69] found a similar relationship between SOM and plant available water.

The extensive studies by Minasny & McBratney [45] and Libohova et al. [69], although insightful, do not resolve the uncertainty surrounding the relationship between SOM and plant available water. The meta-analytic methods in both studies fail to distinguish organic matter additions due to exogenous inputs (e.g., sludge or manure) from organic matter added by living plants and plant residues. Furthermore, neither study spoke to the relative impacts of short-term versus long-term organic matter additions. Soil water content increases most when exogenous organic matter inputs, living plants, and soil fungi are simultaneously present [87], suggesting that analyses that only investigate exogenous organic matter inputs may fail to fully account for PAW contributions from other types of organic matter. A continent-scale study by Bagnall et al. [88] showed that PAW increases attributable to organic carbon were more than double previous estimates made in smaller-scale studies, with non-calcareous soils showing the greatest organic carbon responses.

## 6. Exploring Interactions between Texture, Organic Matter, Structure, and Water Retention

The physical and chemical diversity that exists among SOM pools renders general statements about the relationships between SOM, texture, soil structure, and water retention overly broad with respect to advancing our understanding of soil carbon dynamics across landscapes. Lavallee et al. [89] recommend separating SOM into POM and mineral-associated organic matter, given the functionally distinct features of POM, including its formation and persistence. Whereas total SOM and POM increase with cover crop use [31], management effects on organic carbon storage can be inconsistent across soil textures [38]. These inconsistencies highlight the need for additional research into interactions between organic carbon, soil texture, and soil structure as they relate to soil water retention.

Although texture is the dominant factor influencing soil-water properties, the interactions between texture, SOM, soil structure, and land management practices can significantly affect, and be affected by, soil hydrological properties [90]. The importance of these interactions is also apparent in processes governing soil aggregation. A bimodal distribution of soil pore spaces is common, regardless of texture, with the sole exception being pure sands [91]; soil water retention characteristics, in general, are best described using bimodal functions [92]. An increase in the relative percentage of soil macroaggregates can therefore act to increase the relative abundance of macropore spaces, with concomitant increases in capillary water storage. Living plant roots act to promote soil aggregation [39] and create macropore spaces [93]. Cover crops essentially serve to prolong the presence of living roots in the root zone and therefore have the potential to influence soil water retention through alteration of the soil physical environment.

Landscape management plays a consequential role shaping the soil physical environment. Mollic soils intensely managed under perenniality and ruminant animal agriculture can have substantially more SOC accrual than systems that rely solely on crop diversity and conservation tillage [94], suggesting that grazing can play a significant role in SOC accrual.

Tillage also plays a crucial role in SOC dynamics by reducing the amount of macroaggregates relative to microaggregates, which is significant given that macroaggregates contain up to 30% more carbon and nitrogen per unit volume than microaggregates [95].

No-tillage systems generally have increased aggregation and SOM in the root zone relative to conventional tillage systems. Tillage is a major cause of reduced stability and number of soil aggregates when native ecosystems are converted to agriculture [96]. Soil aggregates act to physically protect SOM from decomposition, as evidenced by a spike in SOM mineralization after aggregate disruption [97]. Six et al. [96] present a theoretical model to explain the process of soil aggregation whereby: (i) Fresh plant residues become intra-aggregate POM, which promotes microbial activity by serving as a carbon energy source. This in turn promotes the production of microbe-derived binding agents (e.g., extracellular polymeric substances) that hold soil particles together. (ii) Intra-aggregate POM transitions from coarse to finer particles as it decomposes. (iii) Microaggregates begin to form within macroaggregates. Eventually the binding agents that hold macroaggregates together weaken, releasing the microaggregates that are held inside. (iv) The released microaggregates serve as the foundation on which subsequent macroaggregates are built.

Soil organic carbon can show a moderate positive correlation with soil aggregate stability and soil water content at the surface layer [98], suggesting that soil aggregation may play a role in increasing SOC and by extension soil water content. However, the extent to which aggregated structures have an impact on soil water retention through either direct alteration of pore size distribution or an indirect impact on soil water content through increased SOM is not fully understood. Although increased root diversity can improve soil structure and enhanced SOC [39], interactions between soil structure, cover crop management, SOC, and soil water are seldom within the ambit of an individual study. A circumspect analysis of the current body of literature suggests that cover crop roots can serve to improve both soil carbon and soil water properties through enhanced soil aggregation; however, the need for a more thorough understanding of the interactions between these components remains. Aggregation and SOM often have negligible impacts on soil water content at low matric potentials; however, SOM can modestly increase water retention at high matric potentials [99], with increases in water retention at higher matric potentials being attributable, in part, to enhanced soil aggregation [65]. Water stable aggregates and POM have been shown to account for the majority of SOC storage, 60% and 20%, respectively [71], suggesting that a comparison of intra-aggregate POM and unprotected POM may provide a useful metric to assess the extent to which POM and water-stable aggregates influence soil water retention.

### 7. Conclusions

Sustained agronomic productivity in temperate rain-fed cropping systems, and concomitantly sustained human health and prosperity, will hinge on the development and deployment of improved soil carbon and soil water management strategies. Bossio et al. [100] offer an apt and succinct summation of this challenge: "every land use decision is a water use decision". Cover crops have the potential to be an effective and practical soil carbon and water management tool in temperate rain-fed cropping systems; however, gaps in our understanding of the spatial and temporal effects of cover crops on soil-water properties, particularly as they relate to interactions between management, organic matter, and the soil physical environment, must be addressed in order to effectuate improved in-field water management strategies. Advancing our understanding of these issues is of urgent concern, and whereas the challenges to filling these gaps in our collective knowledge are not insignificant, they do present substantial opportunities for further research at both mechanistic and landscape-system scales.

**Funding:** This research was funded by Iowa Water Center.

**Data Availability Statement:** Not applicable.

**Acknowledgments:** I would like to especially acknowledge Tom Sauer and Rick Cruse for being so generous with their time and for their thoughtful feedback. I would also like to extend my gratitude to the Iowa State University Department of Agronomy, particularly Kendall Lamkey for providing robust administrative support. I would also like to extend my heartfelt thanks to Thelma Harding at the Iowa State University Graduate College for her tireless efforts to ensure continued administrative and financial support. This publication was made possible through generous financial contributions from the Iowa Water Center, the Alliance for Graduate Education and the Professoriate, and the Graduate Minority Assistantship Program.

**Conflicts of Interest:** The authors declare no conflict of interest.

## Abbreviations

| | |
|---|---|
| EPS | extracellular polymeric substances |
| FC | field capacity |
| PAW | plant available water |
| POM | particulate organic matter |
| POXC | permanganate oxidizable carbon |
| PWP | permanent wilting point |
| SOC | soil organic carbon |
| SOM | soil organic matter |

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
