# Peer review of "Challenges and Opportunities for Cover Crop Mediated Soil Water Use Efficiency Enhancements in Temperate Rain-Fed Cropping Systems: A Review"

_land, doi:10.3390/land12050988_

Round 1

Reviewer 1 Report

Brief summary

The present paper discusses very actual and crucial questions concerning soil carbon and soil water use efficiency in rain-fed cropping systems. Sustainable agricultural management practices are vital for feeding the growing humanity midst the challenges of climate change. Applying cover crops in rain-fed cropping systems is an effective and spreading soil management practice. The topic of the paper is thus very actual. The content of the article is well and logically structured. It gives a good overview and summarises the challenges and opportunities of using cover crops from the context of soil water use efficiency and carbon storage properly. However, review of the recent literature is not complete. Filling this gap could make this paper a good summary of the topic given in its title.

General concept comments

The review is clear, comprehensive and of relevance to the field. Gaps in knowledge are identified.

Review published recently and, if yes, is this current review still relevant and of interest to the scientific community?

The cited references are not mostly recent publications (within the last 5 years) but are relevant. Recent citations are omitted. It does not include an excessive number of self-citations.

Statements and conclusions are drawn coherent and supported by the listed citations.

Figure and table are appropriate, and they show properly the data. They are easy to interpret and understand.

Specific comments

148                  Figure 1 is referred here but the figure itself is in 330. It should be placed closer.

168                  The expression “disproportionately” seems for me not to be the best choice here. I would suggest using „extraordinarily” or „exceptionally”.

358                  “While some studies…” indicates that there are more studies like that. In this case more studies should be cited. While there is only one study with the cited result, the phrase “some studies” should be avoided.

359                  “other studies”: see the opinion for 358

Reviewer 2 Report

Dear author, 

The current manuscript provides helpful information and represents a valuable contribution to an important subject. The paper is generally well-written and structured. The authors provided background information and tried to encompass all relevant references in the introduction. Each section was well written. The overall quality of this manuscript is very good. However, the author could improve the quality and content of the manuscript by providing more background parts with current scenarios, for example, the effect of climate change on crop production, and improve soil quality, including physical, biological, and chemical. My most significant concern is that the author has used outdated references in most of the session, which must be improved. Please updates with more recent citations, for example, line 149-156, 186-200, up to 236, and so on. If you are interested, please refer following papers, published recently, for the background part and soil quality improvement with respect to climate change. 

https://doi.org/10.3390/earth3010004

https://www.frontiersin.org/articles/10.3389/fenvs.2023.1059449/full

Thank you. 

Reviewer 3 Report

A report for: Challenges And Opportunities for Cover Crop Mediated Soil Water Use Efficiency Enhancements in Rain-Fed Cropping Systems: A Review

It is an interesting paper in wich authors review details reghrading challenges and opportunities for utilizing cover crop management to enhance soil carbon stocks and soil water use efficiency in rain-fed cropping system. However, in the abstract a series of generalities is made, but the main achievements of the work do not appear; therefore the abstract must be revised and rewritten again. Although the study could be useful for a certain group of scientists, the manuscript needs improvements before publications.

-Line 123. To date, few studies have investigated cover crops effects on soil water use efficiency in rain-fed cropping systems. And the authors that contribute?

- I believe that there is a wide range of articles that have not been cited. 

- Some map, diagram or illustrative figure should be provided. 

In my opinion the authors should clearly indicate the scientific novelty of the described research. When analyzing the literature, they should indicate what elements of their research contribute to knowledge.

Reviewer 4 Report

Dear Author,
The suggestions are advised below for improving the review manuscript. Pls. go through:
1. Abstract-It is out of context and needs to rewrite again based on your title.
2. Introduction-Not written
3. Tables- Do not use data from the paper directly to write a paper. You can use part of it/change the presentation style and then quote the references.
4. Review should be meta-analyzed/compilation of several pieces of information/literature.
5. Add some good figures for better understanding.
Overall, improve the quality and language of the manuscript.
Regards,

Round 2

Reviewer 1 Report

Dear Author,

I only have one comment which is not very important but since I have noticed it, I tell it:

·         Page 14: “Combined biochar and manure applications may actually serve to immobilize nutrients in medium-textured soils under some conditions (Singh et al. 2020).

Being not a native speaker, I may be wrong but “under some conditions” sounds for me not very appropriate here. I would suggest either to describe shortly the conditions or to use “under certain conditions”.

Author Response

Thank you for the thoughtful feedback, your point is well taken. 

I have removed the words "under certain conditions" as the findings of the reference paper are relevant to most crop production situations. 

Reviewer 2 Report

Dear authors,

Thank you very much for addressing most of my comments and concerns. The manuscript has significantly been improved.

Thanks. 

Author Response

Thank You for the thoughtful feedback and suggestions.

Reviewer 3 Report

I believe that the authors have done a good job, following the reviewers' instructions, and the quality of the work has increased significantly for it to be published in Land.

Author Response

Thank you for the thoughtful feedback and suggestions.